# Application of the Unified Protocol for a Japanese Patient with Post-Traumatic Stress Disorder and Multiple Comorbidities: A Single-Case Study

**DOI:** 10.3390/ijerph182111644

**Published:** 2021-11-05

**Authors:** Noriko Kato, Masaya Ito, Yutaka J. Matsuoka, Masaru Horikoshi, Yutaka Ono

**Affiliations:** 1Department of Neuropsychiatry, Keio University School of Medicine, Tokyo 160-8582, Japan; norikokato@keio.jp; 2National Center for Cognitive Behavior Therapy and Research, National Center of Neurology and Psychiatry, Tokyo 187-8551, Japan; mhorikoshi@ncnp.go.jp; 3National Hospital Organization Disaster Medical Center, Department of Psychiatry, Tokyo 190-0014, Japan; matsuoka-psy@umin.ac.jp; 4Ono Institute for Cognitive Behavior Therapy, Tokyo 102-0072, Japan; yutakaon@gmail.com

**Keywords:** Unified Protocol, posttraumatic stress disorder, cognitive behavioral therapy, transdiagnostic intervention, emotional disorders

## Abstract

(1) Background: The efficacy of the Unified Protocol (UP), a transdiagnostic cognitive-behavioral therapy, with trauma-focused exposure has not been sufficiently demonstrated for post-traumatic stress disorder (PTSD) with multiple comorbidities. This study examined the effects of UP treatment with trauma-focused exposure on symptoms of PTSD and comorbidities in a client who was hesitant about exposure. (2) Methods: The client, who had comorbid dysthymia, social anxiety disorder, agoraphobia, and bulimia nervosa, participated in the UP for 20 sessions over 6 months. The principal diagnosis and symptoms of the comorbid disorders were assessed at baseline, post-intervention, and at the 3-month follow-up. This treatment was conducted as part of a clinical study (UMIN000008322). (3) Results: The client showed improvement in the principal diagnosis and symptoms of the comorbid disorders post-intervention compared with baseline and no longer met the diagnostic criteria for any of the disorders. Considerable symptom improvement was observed with imaginal exposure to trauma memories. (4) Conclusions: The UP was an effective alternative treatment for PTSD and symptoms of comorbidities in this client who was hesitant about exposure to traumatic memories, and that the inclusion of trauma-focused exposure provided sufficient therapeutic effects. Further research is needed to examine the generalizability of our findings.

## 1. Introduction

Treatment guidelines and systematic reviews have recommended a number of trauma-focused cognitive-behavioral therapies (CBT) to treat posttraumatic stress disorder (PTSD), based on their efficacy and lower treatment costs [1,2]. Specifically, prolonged exposure therapy [3], cognitive processing therapy [4], and cognitive therapy [5] have all been efficacious in improving PTSD symptoms [1,6,7], with no differences in efficacy found between them. It is recommended that treatment is selected based on the physician’s clinical judgment informed by the issues of treatment access, acceptability, and client preference [7].

In addition to these treatments, transdiagnostic cognitive-behavioral therapies are attracting attention as a new alternative. The Unified Protocol for the Transdiagnostic Treatment of Emotional Disorders (UP) has one of the strongest evidence bases among such therapies [8]. The theoretical foundation of the UP draws on psychopathology, etiology, biological underpinnings, and CBT efficacy studies. The UP conceptualizes a wide range of mental disorders such as depressive and anxiety disorders as “emotional disorders.” It is a theoretically unified cognitive-behavioral treatment program that targets “negative affect,” “neuroticism,” and “emotion dysregulation” as diagnostic factors common to all emotional disorders. It also teaches adaptive emotion regulation skills. The efficacy of the UP for anxiety and depressive disorders has been confirmed through several open-label studies [9,10,11,12,13], randomized controlled trials [14,15,16,17,18,19], and meta-analyses [20,21,22].

The UP also has been shown to be efficacious for PTSD. In particular, the UP with non-trauma-focused exposure has been studied as an alternative therapy for clients with PTSD or other trauma-related psychopathology who do not want to focus on the trauma. A pilot study has examined the effectiveness of the UP delivered in a group format for veterans (*n* = 52) in an outpatient specialty PTSD clinic at a large Veterans Affairs medical center over 16 weeks [23]. In that study, PTSD symptoms, depressive symptoms, and difficulty in emotion regulation all showed significant improvement after treatment. In addition, a randomized controlled trial of the UP for treatment of trauma-related psychopathology found that participants who received the UP had significantly improved the severity of PTSD, anxiety, and depression compared with those who received usual care, and that the treatment effects were maintained at 6-months of follow-up [24].

Those two studies have shown the efficacy of the UP with non-trauma-focused exposure, but also illustrate its limitations. In both of the studies, PTSD symptoms remained to a certain extent even after UP intervention [23,24], and the introduction of trauma-focused psychotherapy after UP with non-trauma-focused exposures has been suggested. These results in those studies are also consistent with the suggestion that applying a transdiagnostic protocol for negative affect in PTSD might not address the dysregulated fear circuitry that is the focus of current CBT treatment approaches [25]. The inclusion of trauma-focused exposure in the UP may improve its effects, but to our knowledge, there has been little research into the effects of UP treatment with trauma-focused exposure.

The UP is a promising treatment for comorbidities that are barriers to the introduction and implementation of CBT for PTSD. Previous studies have shown that PTSD is often accompanied by other mental disorders, such as depression, anxiety, and eating disorders [26,27]. However, treatment selection for PTSD with multiple comorbid disorders or symptoms is not clearly specified in existing guidelines (e.g., National Institute for Health and Clinical Excellence, 2005). Meanwhile, some reports have shown that comorbid depression interferes with treatment response in PTSD [28,29] and some therapists believe that trauma-focused CBT is not appropriate for clients with comorbidities [30]. Furthermore, negative affect, an underlying factor in PTSD and many comorbidities, is thought to limit the efficacy of CBT that focuses on correcting disruptions in fear circuitry; thus, addressing negative affect might improve the efficacy of CBT for PTSD [25]. This point of contention is also consistent with the views of many treatment providers, with research showing that many view skills for managing negative affect as a factor in readiness to receive trauma-focused CBT [31]. The UP has been developed to address the problem of comorbidities frequently observed in depressive, anxiety, obsessive-compulsive, and trauma-related disorders, and may contribute to overcoming these comorbidity issues [8,20]. However, few reports have detailed the UP treatment process for PTSD patients with comorbidities.

This report describes the treatment process of the UP, including imaginal exposure to trauma memories, for a Japanese patient diagnosed with PTSD and other comorbidities, including dysthymia, social anxiety disorder, agoraphobia, self-harm, and bulimia nervosa. Specifically, we discuss, first, how this UP treatment helped her overcome emotional difficulties related to her primary diagnosis and comorbidities and, second, how trauma-focused exposure contributed to her recovery. Personally identifiable information has been changed to protect patient confidentiality. In Japan, only prolonged exposure therapy is covered by insurance, and clinical studies of cognitive processing therapy are ongoing. However, there is little research on therapies for PTSD that assume a variety of symptoms as targets in a transdiagnostic approach such as the UP. Therefore, this case report is also meaningful as the first report on the application of the UP for PTSD in Japan.

## 2. Materials and Methods

A single-case study was carried out with an A-B design and 3 months of follow-up. This treatment was provided as part of an open-label clinical study of the UP for depression and anxiety disorders [11]. This clinical study was registered at the UMIN Clinical Trial Registry (UMIN CTR: UMIN000008322).

### 2.1. Client Information

The client was a 25-year-old married woman who had graduated from high school and was employed as a clerical worker. At age 20, she entered vocational night school to obtain a qualification to advance her career while continuing to work. However, two years before the start of UP therapy, she was hit by a car while crossing the road on her bicycle. She was hospitalized for 2 months for treatment and rehabilitation of a pelvic fracture. Just before the accident, she had passed the first stage of her qualification exam, but the ensuing hospitalization forced her to abandon the second-stage examination. She also experienced considerable difficulty obtaining compensation for her medical expenses and pain and suffering. Repeated close questioning by a male employee of the driver’s insurance company triggered in her a strong fear of men and telephone conversations. After discharge from the hospital, she continued studying but experienced a number of difficulties attending school and preparing for her examination due to re-experiencing symptoms during the commute and in class, fear of other people, difficulty concentrating, and insomnia.

One year later, the first-stage qualification examination fell on the day after the anniversary of the traffic accident. On the anniversary of the traffic accident, she experienced a panic-like attack and was taken to hospital by ambulance for evaluation. The next day, she took the exam but failed due to feelings of intense anxiety during the exam. Moreover, her teacher reproached her for failing. After these events, she was unable to attend vocational school altogether, tending to stay home when not at work. Around this time, various symptoms began to emerge such as bulimia, self-harm, difficulty with anger control, and depressed mood. Over the next year, she experienced considerable discord in personal relations at work due to anger control issues, work challenges attributable to difficulty concentrating, relationship conflicts with her husband, and the narrowing of her life. Her attending physician recommended prolonged exposure therapy [3] 8 months before the start of the UP, but she declined it at that time, fearing imaginal exposure and preferring to continue with pharmacological therapy alone. When the time she started the UP, she was on 10 mg escitalopram and 10 mg propranolol hydrochloride.

When starting the UP, her major complaints included PTSD symptoms. She re-experienced sirens and shouts and being at the scene of the accident; experienced avoidant symptoms such as being unable to pick up the phone, remain in a crowd or on public transportation. She had increased arousal including being unable to control her anger, which caused problems with family and colleagues, and fear of loud noises. She also experienced negative changes in her cognition and mood, including an inability to remember the accident and persistent negative emotions such as fear and guilt. In addition, she reported various symptoms such as strong anxiety concerning strangers, bulimia, dissociation, suicidal thoughts, and self-harm. Two months before starting the UP, her suicidal thoughts were so severe that after arguing with her husband, she attempted suicide by impulsively jumping off a high-rise building but was stopped by her husband.

She had been functioning very well in the years immediately preceding the accident. However, she experienced her parents’ divorce during her childhood and said she could not remember parts of that period. She also disclosed a history of self-harm (e.g., cutting her arms) due to discord with her mother and maladjustment in junior high school.

### 2.2. Case Conceptualization

When introducing the UP to the client, we conceptualized her case from the UP perspective. The client stated that she was shy and withdrawn in early childhood and indicated a predisposition to feeling negative emotions such as tension and anxiety. Furthermore, her parents’ sudden divorce and experience of fleeing from home with her mother when she was an elementary school student could have formed in her a tendency to regard things as unpredictable and uncontrollable. She had responded to negative emotions using maladaptive emotion regulation strategies from childhood. For example, in junior high school, she had used self-harm to avoid relational distress with her mother or friends. She also avoided memories and emotions relating to her parents’ divorce.

After the traffic accident, she avoided trauma-related stimuli and activities such as riding a bicycle, picking up the phone, or remaining in a crowd or on public transportation. This avoidance impeded the normal recovery process and resulted in the onset of PTSD. Additionally, after failing her exam, she began to avoid seeing other people and going out. She became less involved with people other than her husband and mother and began using binge-eating, self-harm, and suicidal ideation to cope with the distress of relational conflicts. Her negative emotions were in response to (a) memories or physical sensations related to a traumatic event and (b) conflicts in interpersonal relationships. The emotion regulation strategies that she used led to her experiencing stronger and more frequent negative emotions. As a result, she developed dysthymia, social anxiety disorder, agoraphobia, self-harm, and bulimia nervosa.

Her PTSD and comorbidities were thought to have developed based on the inherent tendency to experience negative emotions, beliefs about a situation that evoked negative affect, using long-term maladaptive emotion regulation strategies such as avoidance of situations that evoked negative affect, and self-stimulating behaviors to cope with the negative affect. Therefore, the UP-based intervention was expected to help this client as it could address a wide range of negative emotions as well as PTSD symptoms.

### 2.3. Assessment

Assessments were conducted at baseline, post-intervention, and at the 3-month follow-up. Two independent assessors with a doctoral or master’s degree, who were not involved in the patient’s intervention, were responsible for the assessment. 

An assessor evaluated 15 diagnoses based on criteria in DSM-IV and the International Classification of Diseases, 10th revision, Procedure Coding System (ICD-10), and also evaluated suicidality, using the Mini-International Neuropsychiatric Interview (MINI) Japanese version 5.0.0 [32,33]. The 15 diagnoses were major depressive episodes, dysthymia, mania, panic disorder, agoraphobia, social phobia, generalized anxiety disorder, obsessive-compulsive disorder, psychotic disorder, alcohol abuse and dependence, substance abuse and dependence, anorexia, bulimia, PTSD, and antisocial personality disorder. The client met the criteria for PTSD as a principal diagnosis, dysthymia, agoraphobia, social phobia, and bulimia nervosa as comorbidities, and displayed a high risk of suicide at baseline assessment. The principal diagnosis was selected by an independent assessor from among the diagnoses that were applicable to the client and that most strongly impacted her functional impairment. The assessments by MINI at post-intervention and the 3-month follow-up were performed for only the diagnoses meeting the criteria at baseline and for suicidality.

Additionally, the same rater used a structured interview guide for the Hamilton Anxiety Rating Scale to measure global anxiety symptoms (SIGH-A) [34] and the GRID-Hamilton Depression Rating Scale (GRID-HAMD) to measure severity of depression [35,36]. The assessors had received at least 15 h of training on each of SIGH-A and GRID-HAMD and had passed a practical exam involving assessment of a videotaped patient with at least 80% concordance with the gold standard score. The client’s scores of SIGH-A and GRID-HAMD scores at baseline were 33 and 22, respectively, indicating severe symptoms. On self-report scales at baseline, the client scored 76 points on the Impact of Event Scale-Revised, 53 on the Beck Depression Inventory-II (severe) [37], and 23 on the Sheehan Disability Scale [38,39].

The Overall Anxiety Severity and Impairment Scale (OASIS; [40,41,42]) and the Overall Depression Severity and Impairment Scale (ODSIS; [43,44]) were used to assess changes in depressive and anxiety symptoms, and the Positive and Negative Affect Schedule (PANAS) was used to measure the frequency of experiencing positive and negative emotions before each session. Finally, the presence or absence of adverse events was assessed in each session using the Japanese version of the Common Terminology Criteria for Adverse Events (CTCAE v4.0; [45]). The CTCAE defines severity of an adverse event using a Likert scale (Grade 1, mild; Grade 2, moderate; Grade 3, severe or medically significant but not immediately life-threatening; Grade 4, life-threatening consequences; Grade 5, death). A severe adverse event is defined as Grade 3 or higher.

### 2.4. Therapeutic Intervention

The UP comprises eight modules: motivation enhancement; psychoeducation and tracking of emotional experiences; emotional awareness training; cognitive reappraisal; emotional avoidance and emotion-driven behaviors (EDBs); awareness and tolerance of physical sensations; emotional exposures; and relapse prevention. Module 1 focuses on increasing motivation for treatment by overcoming ambivalence toward treatment and increasing self-efficacy. Module 2 provides psychoeducation on the adaptive function of emotions, an introduction to the three-component model of emotions (thoughts, physical sensation/feelings, and behaviors), and reflection on emotional experiences using a framework of antecedents, responses, and consequences. In Module 3, mindfulness exercises are conducted to develop present-focused and non-judgmental awareness of emotional experiences. Module 4 introduces common negative automatic appraisals and promotes cognitive flexibility through the practice of replacing them with alternative appraisals. Module 5 works on identifying emotion avoidance and emotion-driven behaviors and replacing them with more adaptive behaviors. Module 6 involves interoceptive exposures to increase the tolerance of the physical sensations. In Module 7, emotional exposure exercises are practiced to integrate the skills that have been learned and to confront emotions that have been avoided in the past. Module 8 consists of reflecting on all skills and making a plan to prevent relapse. 

The client received 20 sessions in all over 6 months, with each weekly session lasting 60–90 min. The intervention was implemented by a clinical psychologist with a master’s degree and six years of clinical experience who was receiving weekly group supervision by a clinical psychologist with more than 20 years of experience. The intervention was carried out according to the UP workbook and therapist guide [46]. Adherence to intervention contents in each session was confirmed by a second therapist who attended the session or checked that session’s audio recording against the Therapist Adherence Rating Scale for the UP, developed by Barlow et al. [14].

### 2.5. Course of Therapy

The client was given a case formulation summary and it was explained that the UP would focus on trauma after she had developed the necessary emotional regulation skills and that a wide range of issues, not just PTSD symptoms, would be addressed simultaneously. She agreed to start the UP, stating that she could address her relationship with her husband and binge-eating but preferred not to focus on trauma immediately after starting treatment.

#### 2.5.1. Module 1: Motivational Enhancement for Treatment Engagement

This module comprised two exercises: a decision balance exercise to list the pros and cons of both changing and staying the same and a treatment goal-setting exercise whereby the client sets concrete treatment goals and identifies action steps to reach those goals. In the decisional balance exercise, the client expressed strong anxiety about the treatment addressing accident-related matters. However, in the goal-setting exercise, she began to show high motivation for treatment after the following two goals were set: (1) having a good time with her husband just as before the accident and (2) re-entering vocational school to obtain her qualification. In embodying and setting the stage for the treatment goals, she created steps to gradually restart wearing makeup and get on a train by herself, habits she used to have when she and her husband had a good relationship. She was able to perform these activities in small steps on a week-by-week basis.

#### 2.5.2. Module 2: Psychoeducation and Tracking of Emotional Experiences

In this module, psychoeducation on the adaptive function of emotion and the monitoring of emotional experiences is provided based on the three-modal model of emotion, in which emotional experiences are interpreted as having three components: thought, physical sense/feeling, and behavior. Subsequently, emotional experiences are tracked according to an outline of antecedent conditions (A), emotional responses (R), and short-term and long-term consequences (C) to assess whether that emotional response leads to desirable consequences over time.

Through this ARC exercise, the client reported that her behaviors such as binge-eating, thinking about suicide, and complaining to her husband addressed temporal avoidance of negative emotions and did not contribute to her long-term goals. Additionally, she gradually became able to imagine and practice adaptive behaviors in her daily life. Table 1 shows some of the steps that she worked on in relation to her emotional experiences.

#### 2.5.3. Module 3: Emotional Awareness Training

In this module, three types of emotional awareness exercises (i.e., nonjudgmental and present-focused mindfulness, anchoring to the present, and mood induction) were provided to allow the client to practice nonjudgmental and present-focused emotional observation. The nonjudgmental and present-focused mindfulness exercise was especially helpful in reducing frequent feelings of being “dazed” (i.e., dissociation) and regaining the necessary concentration to engage in the session. For this reason, the exercise was performed for a few minutes at the beginning of each subsequent session. Moreover, after the practice of anchoring to the present was introduced, the client was able to reduce the feeling of being dazed in her daily life.

#### 2.5.4. Module 4: Cognitive Appraisal and Reappraisal

In this module, the therapist and client focused on correcting two forms of thinking traps common to emotional disorders: probability overestimation and catastrophizing. Specifically, cognitive reappraisal was conducted to help generate new and different appraisals to contradict probability overestimation and help de-catastrophize, thereby enhancing the client’s flexibility in appraising situations. 

During this module, she had to transfer to a workplace near the accident site. This event temporarily exacerbated her anxiety, but she was able to cope with it by using the cognitive reappraisal skills (Table 2). 

In addition, during this module, her core automatic appraisal, which underlies all her other automatic appraisals, was identified. The downward arrow technique revealed her core automatic appraisal of “I am not a necessary person.” Based on this understanding, she became able to use cognitive reappraisal effectively to address catastrophizing and self-reproaching thoughts in her personal relationships.

#### 2.5.5. Module 5: Emotional Avoidance and Emotion-Driven Behaviors

In this module, the client enlisted her own emotional avoidance strategies and emotion-driven behaviors and practiced alternative behaviors that opposed them. The specific alternative behaviors of the client are shown in Table 1. She listed her emotional avoidance strategies, enabling her to identify examples of behavioral avoidance such as waiting at a green signal to cross the road even after it had turned red and then green again, as well as cognitive avoidance such as purposeful dissociating, shaking off a thought by shaking her head, or eating something to avoid thinking. Moreover, she realized that smiling all the time and repeating “yes” in interpersonal situations was an example of her avoiding the generation of emotions from personal relationships.

#### 2.5.6. Module 6: Awareness and Tolerance of Physical Sensation

In Module 6, interoceptive exposure was performed to increase tolerance of bodily sensations. During the session, the client practiced hyperventilation, breathing through a thin straw, spinning while standing, and running in place. She was assigned homework that primarily involved running in place. Although running was the most distressing for her, it led to increased tolerance for the sensation of her heart beating faster, helping her learn to separate bodily sensations from fear.

#### 2.5.7. Module 7: Emotion Exposure

Emotion exposure refers to the practice of exposing oneself to various internal and external stimuli that generate strong emotional reactions and that are the targets of emotional avoidance. This is considered the most important module in the UP and is situated at the end of treatment as an opportunity to practice all the skills learned up to that point. Table 3 depicts the emotional and situational avoidance hierarchy. The client engaged in both imaginal exposure to painful traumatic memories as well as in vivo exposures (e.g., remaining in crowds, traveling by train to distant places, having telephone conversations, and meeting strangers).

During this module, she attended job interviews and job training at the new office of her current employer. Real-life exposures were planned and performed to fit with her daily schedule and included making a phone call to inquire about a new job, cycling near the site of the accident, going to a job interview on a packed train, and leaving the office with a new coworker and talking to them while making eye contact. In the imaginal exposure, she remembered the scene of the traffic accident as well as the scene of talking by phone with the insurance company employee. She was assigned daily homework to listen at home to the recording of her imaginal exposure from the previous session. During the first four sessions of the emotion exposure module, she could not listen to the recordings. In the 17th session, she discussed with the therapist her avoidance of listening to the recordings and they brainstormed ways to help her complete the homework. She decided when and where to listen to the recordings and where to put the equipment so that she could do her homework every day. After this session, she became able to listen to the tapes, and her anxiety and depression significantly improved.

#### 2.5.8. Module 8: Relapse Prevention

In the final session, the client reported that she had gotten a haircut at a hair salon for the first time in three years, which was the most difficult task on her emotional and situational avoidance hierarchy. She reviewed her treatment progress with the therapist and agreed that her initial complaints had improved. She set long-term goals of reentering vocational school and further improving her relationship with her husband.

### 2.6. Treatment Adherence

She attended all the sessions without canceling. She also worked on her weekly homework assignments, except for listening to the recording of the imaginal exposure, although sometimes she could not complete all of them.

Her therapist’s treatment adherence was checked for 12 sessions using the Therapist Adherence Rating Scale for the UP and was found to be 89.5% (60/67 sessions). No other interventions were found besides those specified in the UP.

## 3. Results

### 3.1. Outcome

In the symptom assessment at the end of treatment, only suicide risk (low) was relevant in the MINI results. Additionally, marked improvement was observed for all scales assessed. At the 3-month follow-up assessment, there was a mild relapse in symptoms, but the improvement in functioning was maintained. Table 4 shows changes in the assessment measure scores.

Figure 1 and Figure 2 show the time course of depression, anxiety, and positive and negative emotions. As shown in Figure 2, the client’s symptoms of depression and anxiety improved gradually after Module 5, where she practiced alternatives to avoidance and emotion-driven behaviors. Changes were observed in her anxiety and depressive symptoms and frequency of positive and negative emotions after the 17th session, when a solution was introduced to help her complete the homework of listening to an imaginal exposure recording. No severe adverse events were observed during treatment or follow-up. The client discontinued her medications due to side effects before the 10th session of the UP.

### 3.2. Client Perspective

The client remarked after the treatment, “Through cognitive behavioral therapy, I came to think that the accident was a part of my life. Although I had bad times, I found out how kind my husband and the people around me have been in supporting me after it happened. I realize that I am a necessary person to them. I’m going to support them in the future.” She reported that among the skills she had learned, a particularly useful one when she feels fear is becoming aware of her breathing for a moment and observing her own reactions and the events around her.

## 4. Discussion

This case report describes the UP treatment process with a Japanese individual diagnosed with PTSD and comorbid disorders. At baseline assessment, she exhibited numerous symptoms, including PTSD, depression, agoraphobia, social anxiety, bulimia nervosa, dissociation, and self-harm in addition to severe functional impairment. After 20 UP sessions, improvement was observed in not only the symptoms of her principal diagnosis but also in the symptoms of all her comorbid disorders and overall functioning. The course in this case suggests that the UP was an effective alternative treatment for this client who was reluctant to be exposed to trauma memories, that training in emotion regulation skills for negative emotions was effective in partially improving symptoms of PTSD and comorbidities and increased readiness for exposure, and that a focus on trauma was needed to achieve a sufficient therapeutic effect. It was also confirmed that the UP for PTSD was adaptable for this Japanese client without the need for major changes from the original version.

The UP structure of learning emotional regulation skills for a wide range of negative emotions prior to exposure was helpful in encouraging the initially reluctant client to try trauma-focused cognitive-behavioral interventions. She initially showed a strong fear of the therapeutic technique of exposure and declined to begin prolonged exposure therapy. Before the UP started, she was told that only after she acquired the necessary emotion regulation skills would exposure be introduced. This treatment structure gave her a sense of security and enabled her to start the treatment. She was also able to address her relationship with her husband and binge-eating early on in her treatment as she had hoped to do, which motivated her to continue treatment. In addition, the fact that her symptoms decreased significantly before the introduction of exposure (a reduction of 5 points on the OASIS and 6 points on the ODSIS) increased her self-efficacy and motivated her to begin emotional exposure. In previous research, the UP for PTSD was positioned as an alternative treatment for clients who do not prefer to focus on trauma, by not implementing imaginal or written exposure to trauma memories [23,24]. However, a UP process that starts with learning emotion regulation skills before including imaginal and written exposure, as used in the present case, might be an option for patients reluctant to focus on trauma memories.

The course in this case demonstrated that even if a patient has a wide range of comorbidities and problems, UP’s core therapeutic principle of theoretical focus on emotions and their regulation can help treat their symptoms simultaneously. In this case especially, the client was able to understand dissociation, self-harm, suicidal thoughts, bulimia (i.e., binge-eating), and social phobia (i.e., avoidance of personal contact) as manifestations of emotional avoidance that consequently led to long-term persistence of her symptoms. She was able to acknowledge this maladaptive pattern by repeatedly monitoring her emotional responses with reference to the three-modal model of emotion and the ARC form and could then make a list of emotional avoidance strategies. The PTSD and comorbid symptoms that developed as avoidance behaviors (bulimia and avoidance of personal contact) were gradually overcome by facing the situations that she had avoided. In this case, symptoms and functioning improved before starting imaginal exposure to trauma memories, which supports the findings of previous studies [23,24] that non-trauma-focused UP can effectively improve symptoms of PTSD and its comorbidities.

On the other hand, the changes in anxiety and depression symptoms and positive and negative affect seen in this case indicate that the intervention components up to Module 6 and situational emotion exposure had limited therapeutic effect, but that exposure to trauma memories, especially listening to recordings of imagery exposure as daily homework, did help to improve her anxiety and depression symptoms, reduced negative affect, and increased positive affect. These results are consistent with reports that a reasonable number of PTSD symptoms may still remain after UP treatment with non-trauma-focused exposure [23,24], and they support the notion that a transdiagnostic protocol that addresses negative affect in PTSD might fail to address dysregulated fear circuitry, which is the focus of treatment in current CBT approaches [25]. The results also support the suggestion that the UP without exposure to trauma memories should be explored for use as a precursor to trauma-focused psychotherapy [23]. In fact, when the UP with in vivo or interoceptive exposures does not produce sufficient therapeutic effect introducing imaginal or written exposure to the trauma memories might improve this effect.

Finally, the results of this case study demonstrate the potential applicability of the UP to PTSD cases in Japan. Although this study is a case report, which limits its generalizability, the results indicate that the UP may be a promising treatment option for Japanese PTSD patients, in addition to prolonged exposure therapy and cognitive processing therapy.

Several limitations must be considered in this case. First, diagnosis was made by a psychiatrist and confirmed using a simple evaluation procedure (i.e., the MINI) and as such is likely less accurate than more highly structured diagnostic scales such as the Structured Clinical Interview for the DSM-IV. Second, this study used a diagnosis based on DSM-IV criteria, which has the limitation of differing from the current DSM-5 based diagnosis. Third, it was not possible to examine the impact of module content on diagnosis-specific symptoms because assessments of the symptoms for the primary diagnosis and comorbidities were not conducted in each session. Fourth, the symptoms of comorbid eating disorders and social phobia were assessed only using the MINI, so the degree of improvement could not be concretely ascertained. Fifth, the follow-up period was short at three months, so the long-term effects could not be evaluated. Sixth, because evaluations based on the therapeutic mechanism were not made in this case, conclusions cannot be made about the difference in treatment mechanisms between trauma-focused and non-trauma-focused exposures. Seventh, additional investigation in larger numbers of cases and with other study designs is needed to examine potential generalizability. Despite these limitations, this case study demonstrates the potential acceptability and efficacy of the UP that includes imaginal exposure to trauma memories in the treatment of clients who are reluctant to be introduced to exposure as well as those with PTSD and comorbid disorders.

## 5. Conclusions

This report describes a UP treatment process for a client diagnosed with PTSD and comorbidities including dysthymia, social anxiety disorder, agoraphobia, self-injurious behavior, and bulimia nervosa. The course in this case indicates that the UP with trauma-focused exposure was an effective alternative treatment for this client who was hesitant about exposure to trauma memories and that training in emotion regulation skills for negative affect partially improved symptoms of PTSD and comorbidities and increased readiness for exposure. Additionally, trauma-focused emotion exposure might be needed to achieve sufficient therapeutic effects. Further research is needed to examine the generalizability of our findings.

## Figures and Tables

**Figure 1 ijerph-18-11644-f001:**
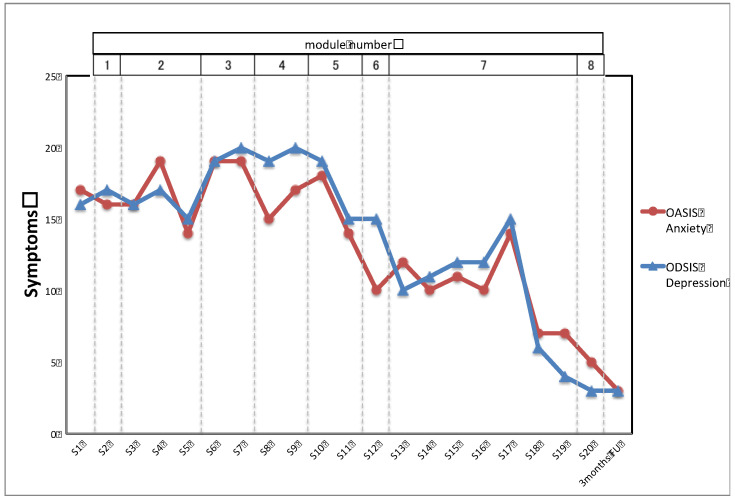
Time course of the client’s depression and anxiety.

**Figure 2 ijerph-18-11644-f002:**
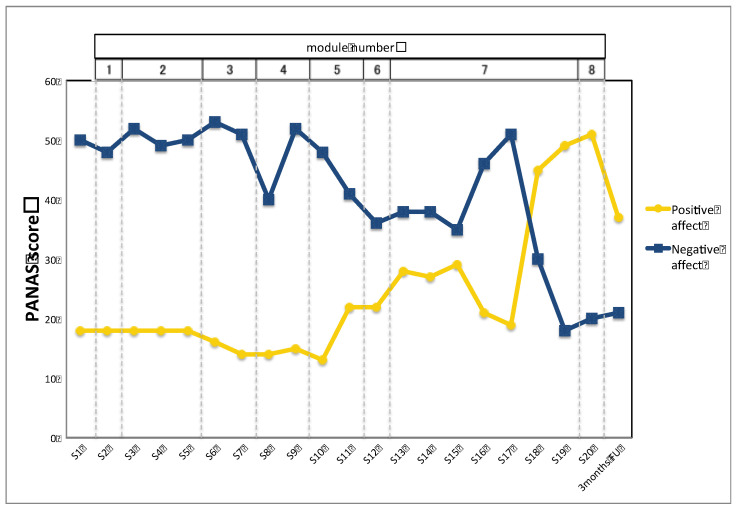
Time course of client’s positive and negative emotions.

**Table 1 ijerph-18-11644-t001:** Monitoring emotions and EDBs in the context of the ARC of emotions.

Antecedents		Response			Consequences
Date/Time	Situation/Trigger	Thoughts	Emotions	Behaviors	What Happened Next?
Monday 6 p.m.	I must see my doctor tomorrow to get a medical certificate concerning the after-effects of the traffic accident.	I don’t know what to do if my doctor rejects my request for a medical certificate.	Fear, anxiety	I went out to get something to binge-eat, but I went home without buying anything.	I felt stressed, but I didn’t feel bad about myself.
Tuesday 6 p.m.	My husband told me he must go to work next Saturday.	Previously, he told me he will take the day off. Could it be that he plans to stay with a woman?	Sadness, anxiety, depression	I complained to him, but I immediately apologized and said thank you for taking the time for us to be together today.	We accepted each other’s situation and talked about plans for the next holiday.
Thursday 4 p.m.	A lawyer told me that the diagnosis of after-effects did not go through the insurance claims process.	Why do so many bad things happen to me?	Sadness, depression, exhaustion, sighing	I got up and filled out a cognitive behavior therapy worksheet.	I could go to the hospital and talk about the problem.
Friday 2 p.m.	I felt lonely because my husband was away from home. I spent my free time watching a movie with lots of blood in it at home alone.	I imagined my own death vividly.	Excited, sensation of palpitations	I stopped watching the movie and started doing housework	I was able to stop thinking about death and made better use of my time.

**Table 2 ijerph-18-11644-t002:** Identifying and Evaluating Automatic Appraisals Form.

Situation Trigger	Automatic Appraisal	Emotions	Thinking Trap	Alternative Appraisal
When crossing at a traffic light.	The traffic light will change before I can cross. I’ll be run over by a car.	Fear, anxiety	Probability overestimation	The traffic light won’t change that quickly. I can run if I need to.
My boss told me a week ago that I must transfer to a new office.	I’ve been abandoned because I did not work hard enough.	Anxiety, depression	Catastrophizing	I finished my job here. I have something to contribute at the new workplace.
My husband worked during a holiday.	He doesn’t want to be with me. We will break up someday.	Anger, depression	Catastrophizing	Sometimes we all have to work on a holiday. While my husband is at work, I can get time for the things I love to do.

**Table 3 ijerph-18-11644-t003:** Emotional and situational avoidance hierarchy.

		Description	Avoid (0–8)	Distress (0–8)
1	Worst	Get a haircut at a hair salon	8	8
2		Ride a bike near the site of the accident	8	8
3		Go for a job interview	7	8
4		Walk alone in a crowd	7	6
5		Ride a crowded train	7	7
6		Pick up a phone call from my lawyer	6	6
7		Call a restaurant to make a reservation	5	6
8		Go to the office near the site of the accident	6	4
9		Make a phone call to a close friend	4	3
10		Pick up a phone call from my mother or husband	0	0

**Table 4 ijerph-18-11644-t004:** Assessment measures at baseline, post-treatment, and follow-up.

Measures	Baseline	Post-Treatment	Three-Month Follow-up
SIGH-A	33	13	17
GRID-HAMD	22	8	13
IES-R	76	10	15
BDI-II	53	11	19
SDS	23	6	4

SIGH-A = Hamilton Anxiety Rating Scale, GRID-HAMD = GRID-Hamilton Depression Rating Scale, IES-R = Impact of Events Scale-Revised, BDI-II = Beck Depression Inventory-II, SDS = Sheehan Disability Scale.

## Data Availability

The data are not publicly available due to privacy protection.

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
