# Peer review of "Application of the Unified Protocol for a Japanese Patient with Post-Traumatic Stress Disorder and Multiple Comorbidities: A Single-Case Study"

_ijerph, 2021, doi:10.3390/ijerph182111644_

Round 1

Reviewer 1 Report

Please, see attached review.

Reviewer 2 Report

The evaluated paper deals with the application of the unified protocol to a case of post-traumatic stress disorder in a young woman with Japanese nationality. 
The case studies are highly interesting for the scientific community and have a high transferability to the general society and applied professionals. The literature review they show is adequate and sufficient. Publication of the manuscript with minor changes is recommended. 

Introduction / Discussion. Within the title the words "Japanese patients" take on special relevance, but this variable is not alluded to at any point in the introduction or discussion. It may be of great interest to readers to know whether or not there are differences in the efficacy of accepted treatments for posttraumatic stress disorder depending on the culture of the patients. It is recommended that the authors include a section alluding to the efficacy of the treatments proposed by the scientific community for posttraumatic stress disorder as a function of the person's culture of origin, and if there is evidence on the use of this protocol for similar cases in samples from other cultures, it would also be of interest. 

On the other hand, the explanation of the protocol and its application is clear and aids reading. But there is a "problem" throughout the text regarding patient assessment that needs to be explained and justified. I understand that a long time may pass between data collection and subsequent preparation for publication, but it is necessary to justify why the authors evaluate the patient with the outdated criteria of the DSM-IV manual (it has not been used for 21 years), and do not use the more updated DSM-V version. Furthermore, if there is a serious reason for this (e.g., that the evaluation and treatment were done while these manuals were in force, which would be strange, since the DSM-IV version was in force from 1994 to 2000, then the DSM-IV.TR version was used between 2000 and 2013, and since 2013 we have the DSM-V version), it is necessary to explain whether the evaluation of the patient with the symptoms she presented, the same diagnosis would be obtained using the current manual. 
